# Unveiling dynamic metabolic signatures in human induced pluripotent and neural stem cells

João Vasconcelos e Sá[1,2‡], Daniel Simão[1,2‡], Ana P. Terrasso[1,2], Marta M. Silva[1,2], Catarina Brito[1,2], Inês A. Isidro[1,2], Paula M. Alves[1,2]*, Manuel J. T. Carrondo[1,2]*

1 iBET, Instituto de Biologia Experimental e Tecnológica, Oeiras, Portugal, 2 Instituto de Tecnologia Química e Biológica António Xavier, Universidade Nova de Lisboa, Oeiras, Portugal

‡ These authors share first authorship on this work.
* marques@ibet.pt (PMA); mjtc@ibet.pt (MJTC)

**Data Availability Statement:** Data are available in the BioStudies database (http://www.ebi.ac.uk/biostudies) under accession number S-BSST375.

## Abstract

Metabolism plays an essential role in cell fate decisions. However, the methods used for metabolic characterization and for finding potential metabolic regulators are still based on characterizing cellular metabolic steady-state which is dependent on the extracellular environment. In this work, we hypothesized that the response dynamics of intracellular metabolic pools to extracellular stimuli is controlled in a cell type-specific manner. We applied principles of process dynamics and control to human induced pluripotent stem cells (hiPSC) and human neural stem cells (hNSC) subjected to a sudden extracellular glutamine step. The fold-changes of steady-states and the transient profiles of metabolic pools revealed that dynamic responses were reproducible and cell type-specific. Importantly, many amino acids had conserved dynamics and readjusted their steady state concentration in response to the increased glutamine influx. Overall, we propose a novel methodology for systematic metabolic characterization and identification of potential metabolic regulators.

## Author summary

Metabolism is no longer considered simply a consequence of cellular regulation, as it can trigger regulatory signals that modulate key cell fate decisions such as proliferation, differentiation and death. To be able to maintain homeostasis, cells regulate and control the metabolic state within defined boundaries. Therefore, to prevent phenotypic changes, key metabolites must be subjected to tight control. Identification of the most tightly regulated metabolites in cellular systems has been hampered by limitations of current methodologies, which typically focus on steady-state data, overlooking transient dynamics and potential regulators. We proposed an innovative approach to face this challenge, by exploring the dynamic response of metabolic pools of human stem cells to a sudden metabolic perturbation. Our data suggests that intracellular metabolic pools respond to extracellular changes in a controlled and cell type-specific manner. This approach might also contribute to systematically uncover potential key metabolic regulators involved in cell

**Funding:** JVS acknowledges his PhD fellowship (PD/BD/52474/2014) to Fundação para a Ciência e a Tecnologia (FCT), Portugal (https://www.fct.pt/). This work was supported by iNOVA4Health UID/Multi/04462/2013), a program co-funded by FCT/Ministério da Ciência, Tecnologia e do Ensino Superior and by FEDER under the PT2020 Partnership Agreement (http://www.inova4health.com/). The funders had no role in study design, data collection and analysis, decision to publish, or preparation of the manuscript.

**Competing interests:** The authors declare that no competing interests exist.

fate decisions, which can be translated not only into advances in stem cell manufacturing and disease understanding but ultimately help unveil new therapeutic solutions.

## Introduction

Over the last decade, the paradigm of metabolism being simply the engine for metabolic constituents has been dramatically shifted. Metabolism generates regulatory responses [1] that affect all other molecular levels, from epigenome to proteome, through diverse action mechanisms (e.g. DNA methylation, histone acetylation) [2–4]. Those regulatory responses modulate key cell decisions such as proliferation, differentiation and death [5]. From this relationship between metabolism and cell fate, it follows that each cell type may have a unique metabolic phenotype.

Inspecting the activity of metabolic pathways is an appealing approach for cell characterization. This approach can pose significant limitations, as recent literature has shown that cells not only fine-tune their intracellular fluxes according to momentaneous needs [6] but can also change their sources of carbon and nitrogen [7]. For instance, it was found that asparagine uptake in mammalian cells preferentially occurs in a glutamine-deficient environment and does not occur when glutamine is present [7].

In an attempt to better characterize metabolic cell status, several studies have explored intracellular metabolic pool quantification. However, dependency of intracellular metabolic pools on extracellular environment was also observed [6,8]. Furthermore, metabolic pools involved in epigenetic regulation are usually identified by looking into substrates of epigenetic enzymes [9–11]. This has the shortcoming of leaving out metabolites that, although apparently unrelated to those enzymes, could influence them due to the complex dynamics of metabolic and regulatory networks. Another limitation of stationary quantitative approaches is that potentially relevant metabolites might be masked if the response to a change in conditions is transient and the metabolite concentration returns to the same levels of the previous cell state [9]. Consequently, not only are the methods used for characterization of cell metabolism dependent on extrinsic factors but can also lead to limited deductions as these are typically based on steady-state data, meaning that transient dynamics and potential regulators are not experimentally observed.

In this work, we hypothesized that cells must be able to maintain homeostasis and stable conditions to prevent small environmental variations from causing substantial changes in cell phenotype. Therefore, the metabolic phenotype of a cell is displayed by the dynamics of metabolite pools and consequently metabolites with a very efficient or robust control of their concentration are potentially key for the cell homeostasis. Observing pool dynamics instead of flux dynamics should be more relevant for accurate cell characterization as pools directly affect several molecular levels such as proteins, DNA and histones which in turn influence cell fate [3,4]. In order to test our hypothesis, two human induced pluripotent stem cell (hiPSC) lines and two human neural stem cell (hNSC) lines were exposed to a step increase in extracellular glutamine concentration. With this challenge, the intracellular dynamic profiles were determined for up to 201 metabolites, covering most of the central carbon metabolism and lipidic pathways. The dynamic profiles were compared between hiPSC and hNSC using process dynamics and control concepts. This approach allowed for the identification of metabolic dynamics conserved and unique for each cell type. Overall, we propose an unbiased and systematic methodology to characterize cells metabolic signatures and to identify potential metabolic pools involved in cell fate decision.

## Results

### Glutamine perturbation experiments of stem cells in stirred-Tank bioreactors

For time-series metabolomics, two cell lines of hiPSC and two cell lines of hiPSC-derived neural stem cells (hNSC) were used (see Materials and Methods). Considering the risk of finding false differences due to cell origin and not due to cell phenotype, one of the hNSC lines used was derived from one of the hiPSC lines (hiPSC 1 and hNSC 1). Cells were cultured as spheroids in stirred-tank bioreactors, displaying high cell viability (Fig 1A). This culture system presents several advantages considering our experimental design, as it allows for a fast and multiple sampling while providing controlled conditions (temperature, pH and $pO_2$) that maximize the biological reproducibility between replicates [12]. Cell spheroids of hiPSC and hNSC maintained their phenotype features for the 3 days of the experiment. In hiPSC cultures, over 95% of cells were positive for pluripotency surface markers Tra-1-60 and SSEA4 (Fig 1B). The neural progenitor markers nestin, SOX2 and vimentin were detected in hNSC cultures, with rare neuronal βIII-tubulin-positive cells (Fig 1C).

Glutamine, a metabolite critical for hPSC [13] and hNSC survival [14–16], was chosen for the extracellular perturbations. Indeed, this amino acid, a versatile donor of nitrogen and carbon atoms for diverse biosynthetic reactions, is preferentially consumed by proliferating mammalian cells in comparison with other amino acids [7]. Nucleotides, non-essential amino acids and the anaplerotic substrates of the TCA cycle are all major biosynthetic products of glutamine in human cells [17]. The intensity of the glutamine step is also important. On one hand, a low intensity could conceal metabolic differences as these would come close to error values of technical sampling replicates. On the other hand, a too high intensity could eventually cause an irreversible homeostatic disruption of intracellular metabolic pools to which cells could not naturally respond and possibly cause phenotypic changes such that metabolic adaptation would not be seen, only the outcome of an uncontrollable disruption. Studies on *E.coli* with glucose steps used an increase of extracellular concentration from 10 to 35 fold [18–20]. However, with glucose being the initial metabolite of the highly active metabolic pathway of glycolysis, cell dynamics might be more robust to glucose steps than to glutamine steps, despite glutaminolysis being also an important and active metabolic pathway for hPSC [13] and hNSC [14–16]. The glutamine concentration after the perturbation step was set to 15 mM, i.e., a step increase of at least 6 times over the initial glutamine concentrations of 2.5 mM, which decreased slightly over time due to consumption (S1 Table). The absence of ammonia accumulation after the perturbation step (S1 Fig) corroborates that the final concentration of glutamine is not cytotoxic, as previously reported in murine PSC [21]. Furthermore, the quantity of glutamine added did not alter significantly the osmolarity or the ammonia concentration (S1 Fig). Sampling was done until 2 hours after the glutamine step, as by that time most metabolic pools reached their new steady-state (S2 Fig). Moreover, cell phenotype does not seem to change after glutamine perturbation: pluripotency markers and cell viability of 2D hiPSC cultures have remained unchanged for 72 hours after glutamine perturbation in subsequent experiments.

### Steady-state changes reveal different metabolic phenotypes between hiPSC and hNSC

To study the effects of an extracellular glutamine perturbation step (Fig 1D) in the intracellular metabolic network, a set of 201 metabolites from different metabolic classes were analysed over time: amino acids, biogenic amines, acylcarnitines, phosphatidylcholines,

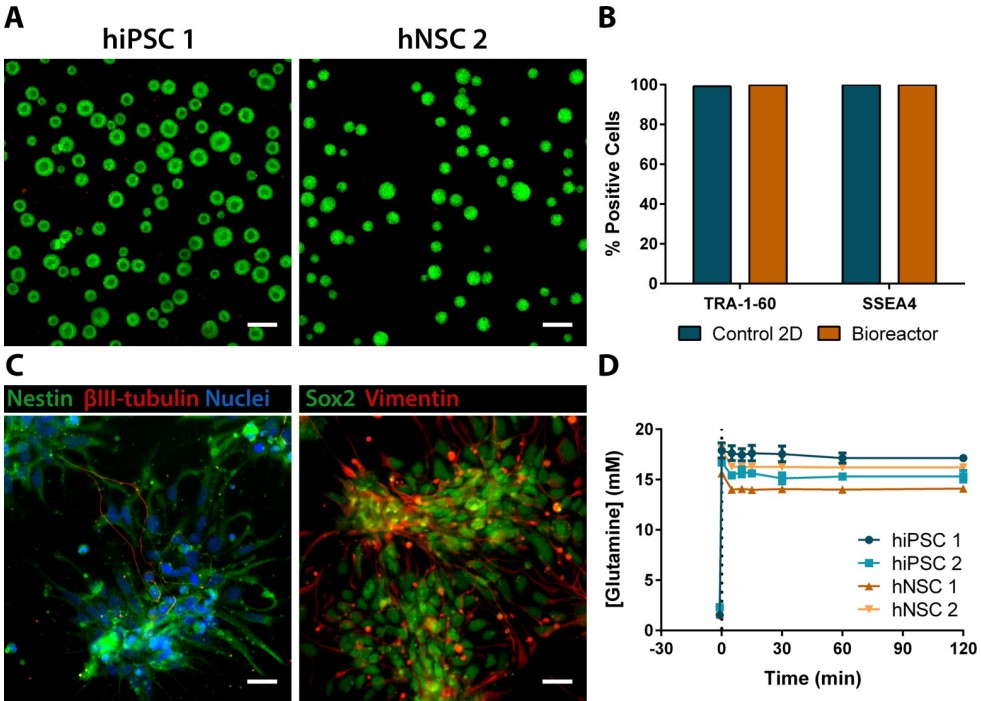

**Fig 1. Perturbation experiments of spheroids of hiPSC and hNSC in controlled bioreactors with a sudden glutamine perturbation step.** (**A**) Viability analysis of hiPSC 1 and hNSC 2 spheroids in bioreactors by staining with FDA (in green, live cells) and PI (in red, non-viable cells). Scale bars, 200 μm. (**B**) Phenotypic analysis of hiPSC 1 by detection of the pluripotency markers TRA-1-60 and SSEA4 by flow cytometry. (**C**) Phenotypic analysis of hNSC 1 for four neural stem cells markers: Nestin, βIII-tubulin, Sox2 and Vimentin by immunofluorescence microscopy. Scale bars, 25 μm. (**D**) Glutamine concentration profile: the extracellular glutamine perturbation step was performed from an initial extracellular concentration up to 2.5 mM until a final concentration of around 15 mM at 0 min. Data are represented as mean of sampling replicates and error bars represent standard deviation.

lysophosphatidylcholines, sphingomyelins and TCA cycle intermediates. For each sample, metabolic pools were quantified, normalized to protein content and then averaged per time-point. Two types of pre-processing operations were performed: (i) at sample level, for removing samples that were considered as mistreated during sample preparation/processing (*i.e.*, presenting metabolite concentrations systematically far from its equivalent replicates) and (ii) at metabolite level, for removing metabolites that could not be quantified with accuracy or at all (*i.e.*, presenting null values or below the limit of detection). For the first operation, samples that caused a relative standard deviation (RSD) over 10% on protein normalized concentration for each metabolite and for each time-point, across all metabolites, were considered outliers and removed from analysis (two outliers in hiPSC 1, hiPSC 2 and hNSC 2 and one outlier in hNSC 1, in a total of 8 time-points x 3 replicates for each cell line). Metabolites that had 5 or more time-points with values under the detection limit or with a RSD on averaged molar quantity per protein above 15% were removed (from 201 measured metabolites for each cell line, 145 metabolites were used in hiPSC 1, 165 in hiPSC 2, 159 in hNSC 1 and 114 in hNSC 2; S3 Table). High variation in average values for different time-points was considered indicative of inadequate extraction or analytical method.

A simple descriptor of steady-state change was determined for each metabolic profile. For each metabolite, changes in steady-states were determined by calculating the ratio of final to initial average molar quantities per protein (fold-change). In order to identify statistically significant changes in steady-states, steady-state values, before the initial glutamine step increase

and 2 hours after, were statistically compared by a two-sample t-test at 5% significance level. If null hypothesis prevailed, the steady-states before and after glutamine step were considered to be the same. Volcano plots of steady-state changes indicate a decrease in the metabolic pools of aminoacids for both hiPSC and hNSC. These plots also suggest a trend towards an increase in central energy metabolic pools for hiPSC, while in hNSC lipids, mainly phosphatidylcholines, appear to be affected in their steady-state value (Fig 2A). These differences were reproducible between the 2 cell lines of each cell type, with Pearson correlation coefficients above 0.7 (Fig 2B). Principal Component Analysis (PCA) showed that most metabolic classes were not clustered in specific regions in the new components space and metabolic classes do not cluster together (S3A Fig). Hierarchical clustering further demonstrates the heterogeneity of metabolic profiles (S3B Fig). Thus, unsupervised analyses indicate that responses to glutamine step were dependent on the metabolite and not on the metabolic class, suggesting that a glutamine perturbation is an adequate experiment to inspect the characteristic metabolic dynamics of each metabolic pool. Overall, these results substantiated our initial hypothesis that steady-state analysis is a relevant method for phenotypic identification.

Mapping the steady-state changes onto a metabolic network exposed the global changes in hiPSC and hNSC metabolism after the glutamine step (Fig 3). Preferred metabolic pathways used to tackle the glutamine influx were indirectly inferred for each cell type. In hiPSC, the glutamine step was absorbed by the lower part of the TCA cycle (from alpha-ketoglutarate to malate). The fueling of this section of the TCA cycle in hiPSC was clearly demonstrated by carbon-labelling in the pivotal work of TeSlaa et al. [9]. On the contrary, in hNSC, the glutamine shock had no effect on increasing the metabolites in the downstream section of the TCA cycle. Instead, the number of lipidic pools for which the steady-state increased was much higher in hNSC than in hiPSC, suggesting an increase in metabolic flux through the upstream section of the TCA cycle. These observations corroborated our previous findings in metabolic flux studies which predicted reductive carboxylation of α-ketoglutarate to fuel fatty acids biosynthesis in NSC [22]. In contrast to the overall response of TCA cycle intermediates and lipids, most amino acid pools decreased their absolute levels (Fig 3). Alanine, arginine and lysine reached far lower steady-states in hNSC. On the other hand, for other amino acids differences were not observed (e.g. threonine) or were observed with fold-changes which were not consistent (e.g. tryptophan).

## Fitting metabolic profiles with a classical process control model exposes conserved transient dynamics

Our data suggests that steady-state changes are cell type-specific, indicative of preferred or active metabolic pathways. We then analysed the transient dynamic profiles of metabolic pools by employing the two liquid surge tanks in series model, typical in the field of process dynamics and control [23]. This model was chosen based on the analogy of metabolic pools as liquid tanks and enzymatic reactions connecting the metabolic pools as tubes connecting liquid tanks. Due to the intricacy and redundancy of the metabolic network that naturally leads to the phenomena of inertia in metabolic pools, a numerator factor was added to the model (see Materials and Methods). The obtained second-order model is still able to fit dynamic models of unknown processes [23], in spite of biological systems incorporating both feedforward and feedback control, along with multiloop and multivariable properties [23,24]. After modelling, mathematical fits with high residual norm were filtered out (see Materials and Methods and S3 Table). A total of 99, 134, 116 and 71 metabolites for hPSC 1, hPSC 2, NSC 1 and NSC 2 respectively, were fitted to a mean fitting error below 5% (S3 Table). The model successfully fitted at least 60% of the metabolites for each cell line. Moreover, it was flexible enough to

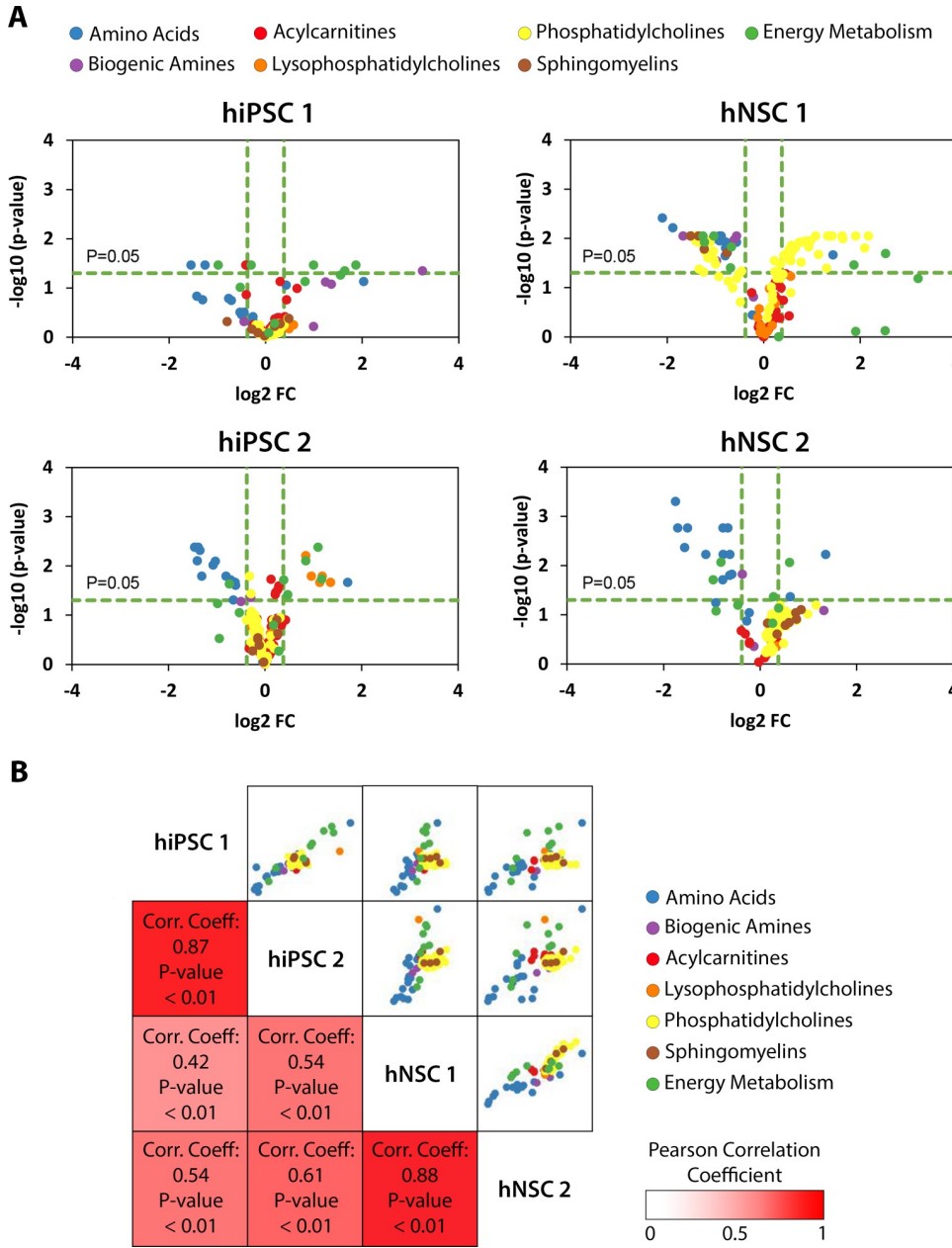

**Fig 2. Steady-state changes of hiPSC and hNSC reveal different and conserved responses to the glutamine step.**
**(A)** Volcano plots of steady-state changes of metabolites. The threshold for Type I error (α) is 0.05, with p-value corrected for multiple testing hypothesis (see Materials and Methods), and for a relevant fold-change is 30% difference from initial steady-state. Positive fold-change means increase of intracellular metabolic pool level after glutamine step.
**(B)** Pearson correlation matrix of fold-changes of steady-states of metabolites. Pearson correlation coefficient spans from -1 to 1 where -1 is perfect negative linear correlation, 1 is perfect positive linear correlation and 0 no linear correlation. Typically, Pearson correlation coefficients between 0.7 and 1 denote a strong positive association.

represent metabolites that reached different steady-states and that presented distinct dynamics such as initial overshoots and oscillations over time (Fig 4). More complex or different types of modelling approaches could be considered. However, to accurately fit the additional parameters of such models, one would need additional sampling points, which in this system would

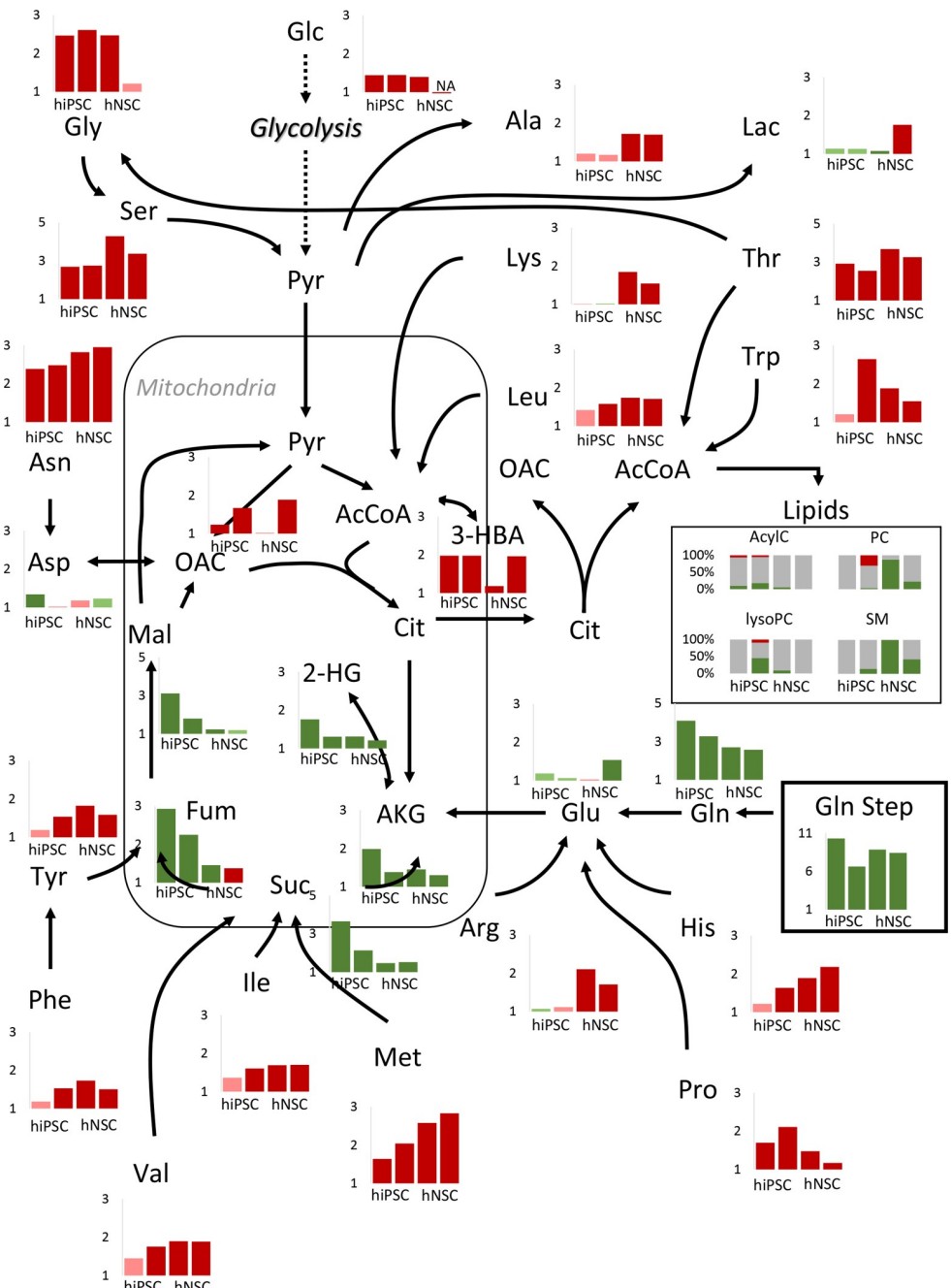

**Fig 3. Steady-state fold-changes mapped onto a metabolic network indicate that global responses of intracelular metabolites to glutamine step are cell-dependent.** In each graph, fold-changes bars are depicted in the order hiPSC 1, hiPSC 2, hNSC 1 and hNSC 2. Green bars denote positive fold-changes, red bars denote negative fold-changes, dark colored bars denote fold-changes statistically significant at 5% significance level, light colored bars for fold-changes which are not statistically significant at 5% significance level. In the lipids box, grey bars denote the percentage of lipids that maintained their steady-state, green bars denote the percentage of lipids that reached higher steady-states and red bars the percentage of lipids that reached lower steady-states, at 5% significance level. AcylC: Acylcarnitines, PC: Phosphatidylcholines, LysoPC: Lysophosphatidylcholines, SM: Sphongomyelins.

be difficult to achieve as we were near the maximum experimentally possible number of samples.

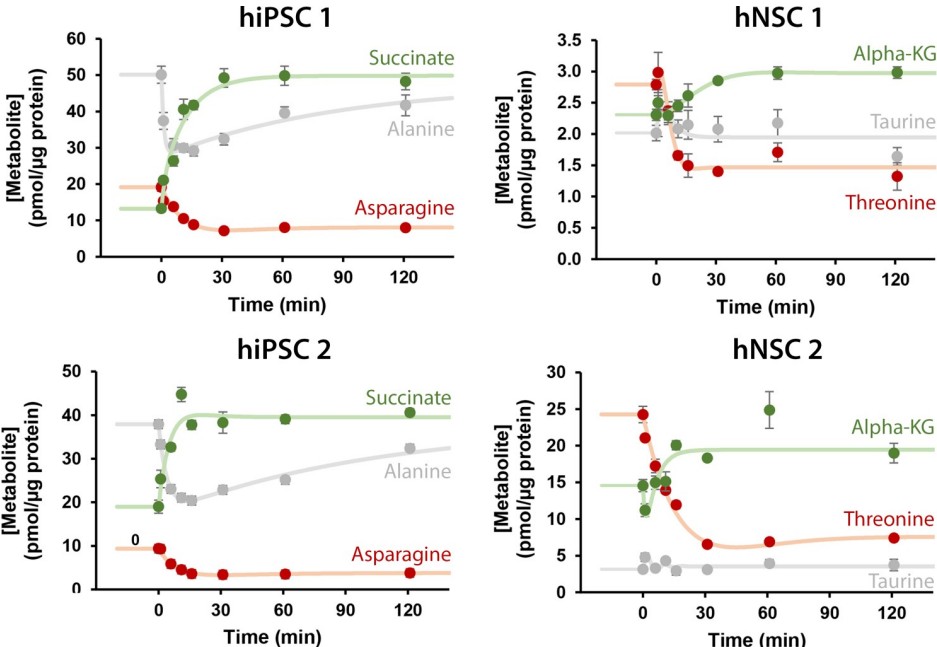

**Fig 4. Mathematical model can simulate different types of dynamic responses.** In green, examples of intracellular metabolites that reach a new and higher steady-state after the glutamine step, in red examples of metabolites that reach a new and lower steady-state and in grey examples of metabolites that keep the same steady-state. Data are represented as mean of sampling replicates and error bars represent standard deviation. Solid lines represent the mathematical fitting to the experimental data.

The modelling of transient dynamic metabolic profiles captures a layer of information that is not present in analysis of dynamics by fold-changes of steady-states. Indeed, using the fold-change of steady-states for discriminating between hiPSC and hNSC increases the quality of the classification model with an area under the curve of 0.79, better than using initial or final steady-states alone (Fig 5A). A classification model is considered good when its area under the curve is above 0.8, where the accuracy of identification is no longer penalized by a high number of false positives. Alternatively, inspecting transient dynamics may contribute to reveal more precisely distinct conserved metabolic features in hiPSC and hNSC (Fig 5B). As depicted for alanine and histidine, transient dynamics reveal metabolic characteristics such as the overall robustness and speed of the response which fold-change of steady-states cannot (Fig 5B). Even when considering the more sophisticated tool of fold-changes of steady-states, it is entirely possible that the same metabolite, in both cell types, can have the same fold-change but in one cell type show oscillations or overshoots while in the other cell type that behaviour is not observed. Therefore, the ability of distinction between cell types is increased when profiling transient dynamics.

## Amino acids show conserved dynamics and readjust their intracellular pools without resorting to oscillations

Following the evidence that metabolic dynamics were generally reproducible between cell lines of the same cell type but not between cell types, metabolites from the same cell type were fit using the same model parameters, except for the parameter process gain, in order to identify which metabolites had conserved dynamics in hiPSC and hNSC (additional details in the Materials and Methods). However, the definition of an acceptable fitting error threshold,

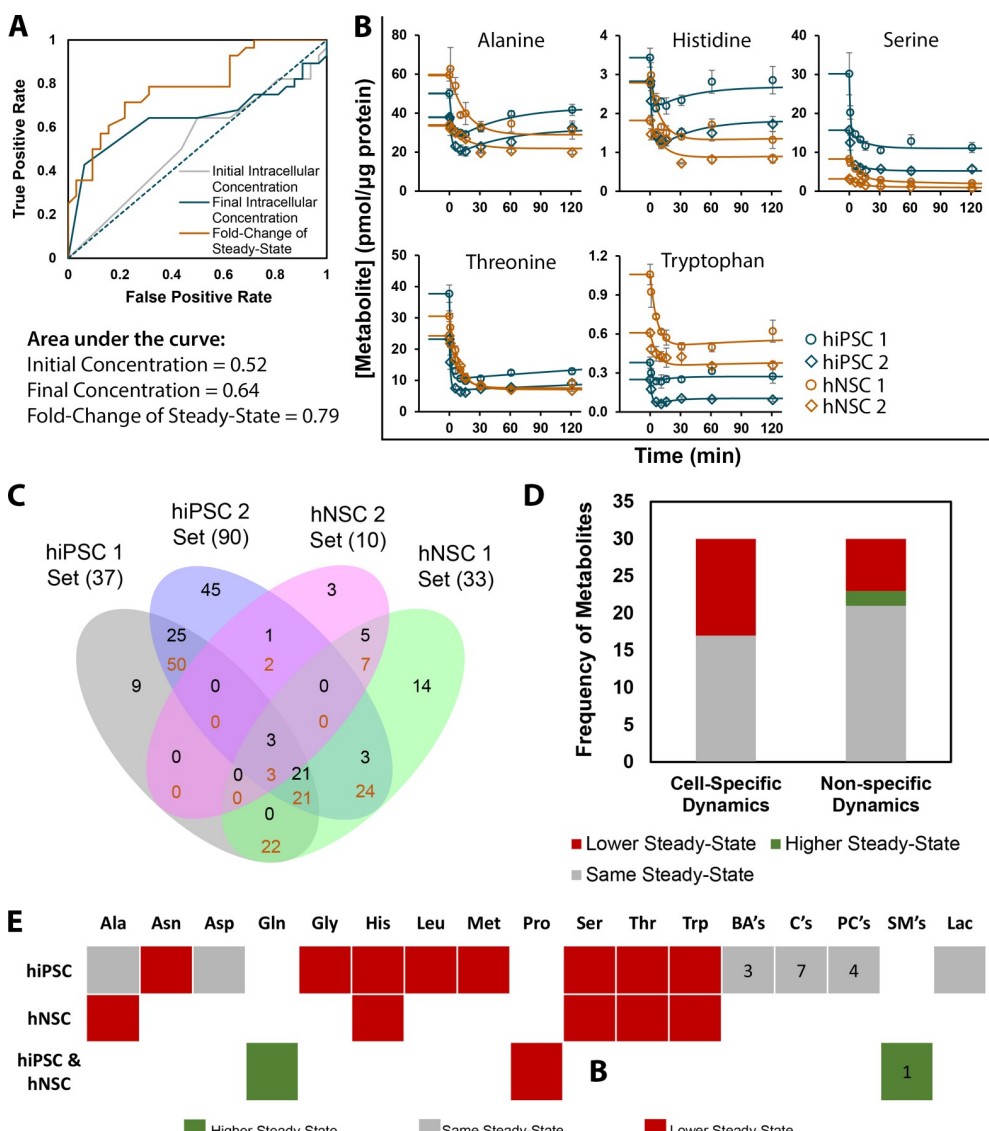

**Fig 5. Identification of metabolites with cell type-specific-dynamics reveals the amino acid class as highly conserved in hiPSC and hNSC and that most of the cell type-specific amino acids decreased their steady-state upon glutamine step increase. (A)** Receiver operating characteristic (ROC) curves show the fraction of metabolic pairs correctly/incorrectly identified as deriving from different cells by the application of three models: one based on comparison of initial intracellular concentration, another based on comparison of final intracellular concentration, and the last one based on the fold-change of steady-state after the glutamine step. **(B)** Metabolic profiles of all metabolites with cell type-specific dynamics in hNSC and their respective cell type-specific dynamics in hiPSC. Experimental points: hiPSC 1 –blue round circles, hiPSC 2 –blue diamonds, hNSC 1 –orange round circles and hNSC 2 –orange diamonds. Adjusted models for cell type-specific dynamics: hiPSC in blue lines and hNSC in orange lines. Data are represented as mean of sampling replicates and error bars represent standard deviation. **(C)** Venn diagram of metabolites with common dynamics. In each intersection, only metabolites with a mean fitting error below 4% are accepted. Black numbers indicate the number of simulated metabolic profiles which fit, specifically to that region and not to any other region with the same or higher number of intersections. Orange numbers indicate the number of all simulated metabolic profiles that fit to that region, regardless of fitting to other regions with the same or higher number of intersections. **(D)** Distribution of simulated metabolic profiles according to the steady-state outcome for metabolites with cell type-specific dynamics and with non-specific dynamics (acceptable fits between at least two cell lines of different cell types). **(E)** Heatmap of metabolites with unique dynamics for hiPSC and for hNSC and of metabolites with dynamics shared by all cells lines, divided in steady-state outcome. Lipids were lumped in classes and the numbers inside its boxes are the number of lipids from that class with dynamics which are cell type-specific or are common to hiPSC and hNSC. BA's: Biogenic Amines; C's: Acylcarnitines; PC's: Phosphatidylcholines; SM's: Sphingomyelins. (See S5 Table for the total list of metabolites with conserved dynamics).

adequate for claiming a different dynamic metabolic response between cells, has to be rationally evaluated.

Therefore, the number of metabolites fitted in all cell lines with acceptable fitting error and included after the pre-filtering step (S3 Table), was graphed over the threshold fitting error (S4A Fig). Each metabolite was fit to all possible combinations for four, three and two cell lines. At a large threshold, all fits of common metabolites belonged to the case of simultaneous fitting to the four cell lines. As the threshold reached to 4% of fitting error, the number of metabolites with fit to the four cells scenario decreases while the fit to hiPSC cell lines and to hNSC cell lines scenario reached their maximum (S4A and S4B Fig). With this threshold for fitting error, the Venn diagram of all metabolic fits showed a relevant amount of metabolites with conserved dynamics, especially for hiPSC (Fig 5C).

When comparing these cell type-specific metabolic profiles with the metabolic profiles that were shared, at least, between two cell lines of hiPSC and hNSC, control characteristics such as settling time and damping coefficient could not discriminate those groups (S5 Fig). Settling time is the time a perturbed process takes to stabilize to a 5% margin of its final steady-state. The damping coefficient relates to the oscillatory behavior. Lower than but close to one means the process is slightly oscillatory in order to reach the region of the new steady-state faster. Above one means that the process is more sluggish and therefore robustness of response is preferred over fastness. This means that damped processes tend to be more stable to unmeasured and unexpected disturbances. Then, for metabolites with unique and shared dynamics, robustness and stable responses seem to be preferred over response speed (S5 Fig).

In terms of steady-state outcome, a substantial fraction of the metabolites with cell type-specific dynamics adjusted their set-point to a lower value after the glutamine step (Fig 5D). This fraction was significantly lower in metabolites with shared dynamics across hiPSC and hNSC. Interestingly, all of these metabolites were amino acids, many with characteristic and distinct dynamics in hiPSC and hNSC (Fig 5B, Fig 5E and S5 Table).

## Discussion

Control of metabolic pools is paramount for cell homeostasis as metabolism has a strong effect on the epigenome, transcriptome, proteome, metabolome and fluxome through varied mechanisms of action. Mathematical modelling of all the interactions of metabolism with other cellular components is still very challenging and appropriate mathematical tools are not yet available. Thus, an understanding of what is the identity of metabolic phenotype when metabolism can adjust itself to environmental changes is slim. Our hypothesis changes the focus from modelling the general to modelling the particular, considering each metabolic pool individually. We resorted to a perturbation experiment where an extracellular glutamine step was applied and dynamic intracellular metabolomics was assessed. The observation of metabolic changes allowed an analysis of the essential and constant metabolic features of the cells, and of the key metabolites that seemed to govern the metabolic response. To our knowledge, this is the first time an experiment of this type is performed in human stem cells to uncover intrinsic dynamics of metabolic pools, while other works have mainly been focused on transition to new metabolic programs. Dynamic metabolomics on adipocytes upon insulin stimulation reported on metabolic rearrangements in central carbon metabolism [25] and a sudden reactive oxygen species stress to *E. coli* unravelled novel allosteric regulations in glycolysis and in pentose phosphate pathway (PPP) [26].

In our study, most metabolites in hiPSC and in hNSC had their most relevant dynamics for 30 min after the glutamine step, similar to what was obtained for metabolites upon insulin stimulation in adipocytes [25]. In *E. coli*, metabolites responded quickly to extracellular

changes and achieved the new steady-state in less than 40 seconds [20,26]. Mammalian cells are larger in size, more compartmentalized and with much slower dynamics. Studies of this type on human cells should be designed to perform intensive sampling during the first 30 minutes after the challenge with time intervals as short as possible. Nevertheless, perturbation steps with metabolites are easier to implement than perturbation of fluxes by overexpressing enzymes [27]. Given the limited time frame, the dynamic response observed in this work should not be regulated at the gene expression level, as protein synthesis rate in mammalian cells is usually in the time frame of hours [28], but rather by regulation of enzyme activity. Future studies focused on the proteome and post-translational modifications could contribute to clarify what are the main responsible molecular regulators Also, correlation with gene expression data (at wider time frames) would be valuable in attempting to understand the downstream impact of the metabolic perturbations and its effect on cell phenotype.

The exploratory approach of this study required broad metabolomics coverage. Different classes of metabolites were quantified, especially from pathways close to the glutamine pathway where the external perturbation was performed: TCA intermediates, amino acids, biogenic amines and lipids. Previous works, in E.coli and human skin cells, sharing related concepts or objectives usually focused on glycolysis and PPP [26,29]. Herein, the focus was on TCA cycle, amino acid metabolism and lipidic pathways, as these pathways have higher probability of being cell type-specific than foundational pathways such as glycolysis and PPP. Indeed, it has been previously reported that TCA intermediates influence hPSC differentiation [9], amino acids and biogenic amines influence the cell fate of hiPSC [30], of T-cells [31] and of oligodendrocyte precursor cells [32]. Lipid levels have been found to be changed between hiPSC and ESC [33] and to promote important signalling for insulin-glucose homeostasis [34]. The Pearson correlation for fold-change of steady-state demonstrated these features were very well conserved in hiPSC and hNSC, independently of the metabolic class. Importantly, this successful correlation demonstrates the potential of simple steady-state fold-change analysis as a method for phenotypic characterization, in contrast with using intracellular metabolic levels for discriminating different cell types, as used recently in cancer cell lines [35]. Moreover, steady-state changes from dynamic analysis allowed for a systemic perspective on the metabolic response of hiPSC and hNSC to the glutamine perturbation. We identified a systemic response towards specific metabolic pathways instead of a uniform flux distribution. This type of behavior has been also observed for adipocytes [25]. Interestingly, our results closely match the data obtained by carbon-labelling experiments, usually more expensive and labor-intensive [9,22]. Thus, the use of perturbation experiments could be developed in the future to benefit metabolic flux studies.

The correlation of metabolic steady-state fold-changes between hiPSC and hNSC was still high and statistically significant. In order to prove with greater confidence that metabolic dynamics are cell type-specific, analysis of transient dynamics was used. The chosen model for fitting was the simplest one that could incorporate inertia in response. In order to get 100% of successful fittings, more complex models (and different types) would have to be used. That would allow us to detect different groups of dynamic response but would require more sampling points, as the number of parameters to be fitted would increase. Nevertheless, the relatively simple model permitted the identification of metabolites with cell type-specific dynamics. Half of them decreased their intracellular metabolic steady-state against one quarter of the metabolites with shared dynamics across cell types. Interestingly, all of those metabolites with cell type-specific dynamics and with decreased steady-state were amino acids. Some of the identified amino acids are known to have important cell regulatory functions. A controlled level of intracellular methionine and of the enzymes involved in its metabolism has been shown to be crucial for maintenance of pluripotency in hiPSC [30]. In a specific subtype of

breast cancer with stem-cell like properties, asparagine has been pointed out as a key factor for governing metastasis [36]. Asparagine has also been shown as regulator of protein synthesis in mammalian cells [7]. Serine supports proliferation of the breast cancer cells [37]. In addition, serine as well as glycine, are involved in one-carbon metabolism that influence epigenetics in human cells [38]. Besides the biological importance of the identification of amino acids as having dynamics which are cell-type specific, this result suggests that targeted approaches for identification of amino acid dynamics may be sufficient for cell identity characterization, instead of the more complex and expensive untargeted metabolomics studies. Moreover, the prevalence of amino acids in metabolic response is advantageous as respective analytical methods are easier, amino acid pool levels tend to be large allowing a simulation fit of dynamic data, and many experimental means are available to the researcher such as label tracers or enzymatic activation/inhibition chemicals.

In the near future, the analysis of dynamics by control parameters such as damping coefficient and settling time might be decisive for ranking the metabolites as potential targets for cell homeostasis regulation. This would tackle another problem concerning classical comparative metabolomics where it is unclear how to accurately identify which molecule among the numerous changed metabolic pools is likely to be the most effective phenotype modulator [39]. The manipulation of effective phenotype modulators to induce cell fate decisions would bring enormous advances in cell therapy and regenerative medicine. After all, reprogramming somatic cells into pluripotent stem cells, expanding stem cells, differentiating stem cells and transdifferentiating cells constitute bioprocesses that are often time-consuming, inefficient and expensive.

Overall, we propose a methodology with considerable specificity for metabolic characterization and for the identification of metabolites characteristic of a cell phenotype by modelling dynamic metabolomics. In this work, we identified metabolic signatures of stemness of hiPSC and of NSC that can potentially be used to solve lingering doubts about differences in phenotype related to cell origins, in the pluripotent and neural stem cell fields. The unbiased nature of the proposed method allows it to be expanded to many other metabolic pathways by performing perturbation steps with different metabolites and by performing more comprehensive untargeted metabolomics, which is increasingly improving its sensitivity and throughput [39]. Coupling these dynamic studies with mathematical modelling in future investigations will lead to a better metabolic understanding on cell regulation and opens an avenue for cell fate manipulation.

## Materials and methods

### Cell culture

Primed hiPSC IMR90-4 (RRID: CVCL_C437) were purchased from WiCell and WTC-11 (RRID: CVCL_Y803) were obtained from The J. David Gladstone Institutes, designated throughout the text by hiPSC 1 and hiPSC 2 respectively. Primed hiPSC were maintained under feeder-free conditions with Matrigel (Corning Matrigel hESC-Qualified Matrix and Corning Matrigel Growth Factor Reduced (GFR) Basement Membrane Matrix, BD Biosciences) and fed daily with mTeSR1 medium (STEMCELL Technologies). Versene (Gibco Life Technologies) and Accutase (STEMCELL Technologies) were used to enzymatically dissociate hiPSCs into single cells for hiPSC 1 and hiPSC 2, respectively. At cell passage, mTeSR1 was also supplemented with 5 μM of ROCK inhibitor Y-27632 (Calbiochem). Complete medium exchange was performed every day. Cells were maintained under humidified atmosphere with 5% $CO_2$, at 37˚C.

hNSC 1 was derived from hiPSC 1 IMR90-4 using a dual SMAD inhibition protocol [40]. Briefly, hiPSC differentiation was induced by supplementing the culture media with 10 μM SB431542 and 1 μM LDN193189 (both from STEMCELL Technologies) for 10 days. hNSC 2 was originally derived from hiPSC line RIi001-A (RRID: CVCL_C888), as previously described [41] and designated throughout the text as hNSC 2. hNSC were expanded in DMEM/F12 (Invitrogen) with Glutamax, N2 and B27 supplements (Invitrogen), 20 μg/mL insulin and 20 ng/mL of bFGF (Peprotech) and of EGF (Sigma). Half of culture media volume was exchanged every other day [41]. hNSC were maintained under humidified atmosphere with 5% $CO_2$ and 3% $O_2$, at 37˚C.

## Stirred-tank bioreactor cultures

hiPSC and hNSC were inoculated in 200 mL of media as single cell suspensions of $0.25x10^6$ cell/mL and $0.4x10^6$ cells/mL, respectively, into software-controlled stirred-tank DASGIP Bioblock bioreactor system (Eppendorf). hiPSC 1, hiPSC 2, hNSC 1 and hNSC 2 were used for these experiments at cell passage number P40, P36, P12 and P34, respectively (four bioreactor runs in total). Bioreactor temperature was set to 37˚C, dissolved oxygen to 15%, pH to 7.4, aeration rate to 0.1 vvm and the agitation rate to a range from 70 to 100 rpm [12,42]. Perfusion was initiated after inoculation and interrupted just before the perturbation experiment. Perfusion rates of hiPSC and hNSC were 1.3 $day^{-1}$ and 0.33 $day^{-1}$, respectively. Cells were allowed to aggregate for 2 to 3 days before performing the perturbation experiment.

## Perturbation experiments and sampling for metabolomics

Before initiating the perturbation experiment, perfusion was interrupted. Glutamine concentration in the culture medium was determined using an YSI 7100 MBS analyser (YSI Life Sciences, Yellow Springs, Ohio USA) offline. The glutamine pulse was induced by adding the required volume of glutamine concentrated solution (L-Glutamine, 200 mM, Gibco) to attain a concentration of 15 mM in the culture media. Changes in osmolarity of the culture medium of bioreactor cultures were later determined using a K-7400S Semi-Micro Osmometer (KNAUER Wissenschaftliche Geräte GmbH, Germany). Sampling was performed before the glutamine step and at several time-points after the step: immediately (0 min), 5 min, 10 min, 15 min, 30 min, 1 h and 2 h after. A sample of 15 mL of culture was collected per time-point sample and distributed equitably in three 50 mL tubes, containing ice-cold PBS to quench cell metabolism, generating three technical replicates that were processed independently. After centrifugation at 300$xg$ for 3 min at 4˚C, a sample of supernatant was stored for later quantification of extracellular glutamine, glucose, lactate and ammonia. The remaining supernatant was discarded and the cell pellet was washed with ice-cold PBS and centrifuged again. The supernatant was removed and a solution of 40:40:20 acetonitrile:methanol:water was added to extract intracellular metabolites from the cell pellet. Sonication was performed to guarantee a complete cell lysis. The extracts were transferred to microcentrifuge tubes and centrifuged at 20000 x g at 0˚C for 15 minutes. Supernatant was collected, snap-frozen in liquid nitrogen and stored at -80˚C until metabolomic analysis. The pellet was also snap-frozen in liquid nitrogen and stored at -80˚C until protein quantification. Protein was dissolved in lysis buffer containing 2% SDS (v/v) and quantified using a Microplate BCA Protein Assay Kit (Thermo Scientific). Ammonia was quantified using the Ammonia Assay Kit (Megazyme).

## Cell viability

Cell viability in spheroids was analysed before the glutamine perturbation experiment by staining the spheroids with fluorescein diacetate (FDA) in PBS (0.02 mg/mL) and propidium iodide

(PI) in PBS (0.002 mg/mL), followed by visualization by fluorescence microscopy using an inverted phase contrast microscope (Leica Microsystems GmbH).

## Flow cytometry

Single-cell suspensions of hiPSC were prepared by Versene/Accutase treatment of cell spheroids. Cell density was determined and $0.5 \times 10^6$ cells were transferred to a microcentrifuge tube, centrifuged at 300$xg$ for 5 min and washed with 2% FBS in PBS. The cells were resuspended in 50 μl of a solution containing the primary antibody: TRA-1-60 (Santa Cruz Biotechnology, sc-21705, dilution 6:100) or SSEA4 (Santa Cruz Biotechnology, sc-21704, dilution 1:10). Cells were incubated with the primary antibody solution for 1 hour at 4˚C, washed with 2% FBS in PBS and centrifuged twice, followed by 30 minutes at 4˚C incubation with AlexaFluor 488 secondary antibodies (Invitrogen, A21042 for TRA-1-60 and A11001 for SSEA4, dilution 1:1000). Cells were washed and centrifuged twice with 2% FBS in PBS, and finally resuspended in 500 uL of 2% FBS in PBS for flow cytometry analysis. Data was collected on a CyFlow Space flow cytometer from Partec. Cells were gated on forward and side scatter dot plots. 10,000 events per sample were acquired and the data were analyzed with FloMax software (version 3.0).

## Immunofluorescence microscopy

hNSC spheroids were plated on sterile glass coverslips inserted on 24-well plates and left for adherence at 37˚C and 5% $CO_2$. Each coverslip containing spheroids were washed once with cold PBS +/+ and then fixed in 500 μL of 4% paraformaldehyde + 4% sucrose in phosphate-buffered saline (PBS) for 20 min at room temperature. Before storage, fixed cells were washed twice with 500 μL PBS. Cells were blocked and permeabilized with 0.2% FSG (Gelatin from cold water fish skin, Sigma, G7765) + 0.1% TritonX-100 in PBS for 20 minutes at room temperature. Primary antibodies were diluted in 0.125% FSG in PBS + 0.1% TritonX-100 and added to fixed spheroids for an incubation of 2 hours at room temperature. Afterwards, cells were washed twice with PBS and incubated with secondary antibodies diluted in 0.125% FSG in PBS for 1 hour and protected from light. Primary and secondary antibodies were used as follows: anti-nestin (Merck Millipore, AB5922), anti-Sox2 (Merck Millipore, AB5603), anti-βIII-tubulin (Merck Millipore, 1:200, MAB1637), AlexaFluor 488 goat anti-rabbit IgG (Invitrogen, A11008), AlexaFluor 594 goat anti-mouse IgG (Invitrogen, A11005). Coverslips were mounted in ProLong Gold antifade reagent with DAPI (Invitrogen, P36935) for staining of cell nuclei. Preparations were visualized on an inverted microscope Leica DMI6000 B (Leica Microsystems). The obtained images were processed using FIJI software [43] and relying solely on linear adjustments.

## Metabolomic analysis of intracellular extracts

Targeted and quantitative metabolomic analysis was performed using the AbsoluteIDQ p180 kit and the Energy Metabolism Assay (Biocrates Life Sciences AG, Innsbruck, Austria). The two assays quantify a total of 201 metabolites from different biological classes, including amino acids, biogenic amines, acylcarnitines, lysophosphatidylcholines, phosphatidylcholines, sphingomyelins and several metabolites of the energy metabolism. For the first assay, analyses were carried out after phenylisothiocyanate (PITC)-derivatization in the presence of internal standards by flow-injection tandem mass spectrometry (FIA-MS/MS, for quantification of acylcarnitines, (lyso-) phosphatidylcholines, sphingomyelins, hexoses) and liquid chromatography-tandem mass spectrometry (LC-MS/MS, for amino acids, biogenic amines) using a SCIEX 4000 QTRAP (SCIEX, Darmstadt, Germany) and a Xevo TQ-S Micro (Waters, Vienna, Austria) instrument with an electrospray ionization (ESI) source. The experimental

metabolomics measurement technique is described in detail by patent US 2007/0004044 (accessible online at http://www.freepatentsonline.com/20070004044.html). For the second assay, after derivatization to their corresponding methoxime-trimethylsilyl (MeOx-TMS) derivatives, energy metabolites were determined by gas chromatography-mass spectrometry (GC-MS) using an Agilent 7890 GC/5975 MSD (Agilent, Santa Clara, USA) system. Pretreated samples were evaporated to complete dryness and subjected to a two-step methoximation-silylation derivatization. N-methyl-N-(trimethylsilyl) trifluoroacetamide (MSTFA) was used as silylation reagent. Split injection was performed and chromatograms were recorded in selected ion monitoring (SIM) mode. External standard calibration curves and ten internal standards were used to calculate concentrations of individual energy metabolites. Data were quantified using the appropriate MS software (Agilent, Masshunter) and imported into Biocrates MetIDQ software for further analysis.

## Data pre-processing and statistical analyses

Absolute metabolic values were normalized by the protein content of the cell pellet for each replicated sample (S2 Table). Metabolites with more than 62.5% of missing values or with coefficients of variation greater than 15% were excluded.

For unsupervised analyses, normalized and averaged metabolic values per time-point were z-scored by subtracting to each value the mean for each metabolite-cell and then dividing by the respective standard-deviation. Principal component analysis and hierarchical clustering was performed in Matlab R2015b (MathWorks, Natick MA) and in Perseus software [44], respectively.

Steady-state fold changes were statistically tested by performing a two-sample t-test, two-sided, assuming the two samples comes from independent random samples from normal distributions with equal means and equal but unknown variances. The Benjamini-Hochberg method was used to correct for multiple testing errors using a false discovery rate of 5% [45]. These fold-changes were log2-transformed for depiction in volcano plots and in the Pearson Correlation matrices. These statistical tests and correlations were performed in Matlab R2015b (MathWorks, Natick MA).

## Dynamic modelling and characterization of parameters

A classical model from process dynamics and control based on two liquid surge tanks placed in series [23] was used for modelling the dynamical metabolic profiles. The specific model, named second order with numerator dynamics, has different equations for two scenarios: one for an underdamped process and another for an overdamped process, displayed below with a complex variable $s$.

$$Overdamped : y'(s) = \frac{KM(\tau_a s + 1)}{(\tau_2 s + 1)(\tau_1 s + 1)} \tag{1}$$

$$Underdamped : y'(s) = \frac{KM(\tau_a s + 1)}{\tau^2 s^2 + 2\zeta \tau s + 1} \tag{2}$$

Metabolic profiles were fit to these two equations by minimization of the sum of squared residuals. The scenario that presented the lowest residual was chosen. The parameter $M$ was calculated based on the concentration of extracellular concentration of glutamine (S1 Table). Fitting of the four other parameters, the steady-state gain K, the numerator coefficient $\tau_A$, the response time $\tau$ and the damping coefficient $\zeta$, was performed in MATLAB using lsqnonlin and nlinfit functions. The former function was used to get a first estimation of model

parameters which would serve as initial parameters to the latter, as the latter function accepts standard-deviations as weights for fitting. Fits with residual norm above 4% were not considered. The fitting of intracellular glutamine for the four cell lines, instantly subjected to the sudden extracellular glutamine step, using the same model parameters except one, display considerable resemblance and very low average fitting error (S6 Fig). So, to tackle randomness variable affecting the quantification of moles of metabolites per protein quantity between time-points, experimental values of hNSC (especially affected by the mentioned random variable) were normalized to the shared glutamine profile by multiplying the ratio of simulated value per experimental value of glutamine at that time-point and for that cell line (for all i, g and y, Met normalized $_{i,\text{cell line } g, \text{ time-point } y}$ = Met $_{i,\text{cell line } g, \text{ time-point } y}$ x (Gln simulated $_{\text{cell line } g, \text{ time-point } y}$ / Gln experimental $_{\text{cell line } g, \text{ time-point } y}$).

The settling time of each fitting curve was determined by finding the time-point after which the metabolic pool value would remain inside a band whose width is equal to ±5% of the final metabolic pool concentration.

The damping coefficient in the overdamped case was calculated directly from the model parameters obtained for each metabolite:

$$In\ Overdamped : \zeta = \frac{\tau_1 + \tau_2}{2\sqrt{\tau_1 \tau_2}}$$

## Supporting information

**S1 Fig. Effect of glutamine steps in hiPSC and hNSC bioreactions on extracellular environment. (A)** Osmolarity of the solution of glutamine used for the perturbation step, of the cell culture media used for hiPSC and hNSC and of the culture media of the four bioreactor cultures immediately after the glutamine perturbation step. Changes in osmolarity after the perturbation step are indicated in percentage on top of each bar. **(B)** Ammonia concentration in bioreactors culture media.
(TIF)

**S2 Fig. Intracellular metabolites reach their steady-state after approximately 2 hours.** Metabolic profiles of alanine and threonine in an experiment covering up to 24 hours after the glutamine step increase demonstrate that 2 hours is usually sufficient for reaching a new metabolic steady-state.
(TIF)

**S3 Fig. Unsupervised analysis of dynamic profiles of intracellular metabolites after glutamine step perturbation.** The dynamic profiles of molar quantities per protein were normalized by a z-score procedure (see Materials and Methods). **(A)** Principal component analysis of metabolic profiles. **(B)** Hierarchical clustering of metabolic profiles. Rows represent the different metabolites, while each column represents one time point (BP–before pulse, 0, 5, 10, 15, 30 min, 1, 2 hours).
(TIF)

**S4 Fig. Selecting the ideal fitting error threshold to allow a confident identification of metabolites with cell-conserved dynamics. (A)** Frequency of fitted metabolites along the threshold of the fitting error, to several combinatorial groups of cells. **(B)** Venn diagram of metabolites, present in all four cell lines, with fits below a 4% error to all cell types. Orange numbers indicate the number of all simulated metabolic profiles that fit to that region, regardless of fitting to other regions with the same or higher number of intersections.
(TIF)

**S5 Fig. Comparison of control-related parameters of simulated metabolic responses between metabolites with cell type-specific dynamics and with shared dynamics across cell types. (A)** Boxplot of settling time of simulated metabolic profiles between cell type-specific and shared dynamics (non-specific). **(B)** Boxplot of damping coefficient of simulated metabolic profiles between cell type-specific and shared dynamics (non-specific).
(TIF)

**S6 Fig. Modelling glutamine dynamic profile for all cell lines using the same model parameters, except of steady-state gain. (A)** Metabolic profile over two hours for each cell line. Experimental points: hiPSC 1—blue round circles, hiPSC 2—blue diamonds, hNSC 1—orange round circles and hNSC 2—orange diamonds. Simulated profiles: hiPSC in blue lines and hNSC in orange lines. Experimental data are represente as mean of sampling replicates and error bars represent standard deviation. **(B)** Parameters used for modeling glutamine profiles. **(C)** Step-response descriptors from glutamine profile modeling for each cell line.
(TIF)

**S1 Table. Step inputs of extracellular glutamine concentration for the different bioreactors.**
(XLSX)

**S2 Table. Complete metabolic quantification dataset for each cell line.**
(XLSX)

**S3 Table. Number of metabolites after each data processing for each cell line.** The "Pre-filtered" step refers to the step where metabolites that had 5 or more time-points with values under the detection limit or with a relative standard deviation on averaged molar quantity per protein above 15%, were discarded. Metabolic profiles were then fitted to an equation model and those with a mean fitting error above 5% were discarded.
(XLSX)

**S4 Table. Model parameters for simulated metabolite profiles of each cell line.**
(XLSX)

**S5 Table. Metabolites with unique dynamics for hiPSC, hNSC and metabolites with dynamics shared by all cells lines, divided in steady-state outcome.** Metabolites which have characteristic dynamics for hiPSC and also have characteristic dynamics for hNSC are underlined.
(XLSX)

## Acknowledgments

We gratefully acknowledge Dr Tomo Šaric (University of Cologne, Germany) for the supply of hNSC 2. We thank Francisca Arez (iBET) for assistance with sample processing. We also thank Rui M.C. Portela (GSK Vaccines) and Marta Abreu Paiva (iBET) for insightful discussions.

## Author Contributions

**Conceptualization:** João Vasconcelos e Sá, Daniel Simão, Inês A. Isidro, Paula M. Alves.

**Data curation:** João Vasconcelos e Sá.

**Formal analysis:** João Vasconcelos e Sá, Daniel Simão.

**Funding acquisition:** Paula M. Alves, Manuel J. T. Carrondo.

**Investigation:** João Vasconcelos e Sá, Daniel Simão, Ana P. Terrasso, Marta M. Silva.

**Methodology:** João Vasconcelos e Sá, Daniel Simão, Ana P. Terrasso, Marta M. Silva, Catarina Brito, Inês A. Isidro.

**Project administration:** João Vasconcelos e Sá, Daniel Simão, Catarina Brito, Inês A. Isidro, Paula M. Alves, Manuel J. T. Carrondo.

**Resources:** Ana P. Terrasso, Marta M. Silva, Catarina Brito, Paula M. Alves, Manuel J. T. Carrondo.

**Software:** João Vasconcelos e Sá.

**Supervision:** Catarina Brito, Inês A. Isidro, Paula M. Alves, Manuel J. T. Carrondo.

**Validation:** João Vasconcelos e Sá, Daniel Simão, Inês A. Isidro.

**Visualization:** João Vasconcelos e Sá, Daniel Simão, Inês A. Isidro.

**Writing – original draft:** João Vasconcelos e Sá.

**Writing – review & editing:** João Vasconcelos e Sá, Daniel Simão, Catarina Brito, Inês A. Isidro, Paula M. Alves, Manuel J. T. Carrondo.

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
