## [Decision Letter · Decision Letter 0]

29 Sep 2019

Dear Dr Alves,

Thank you very much for submitting your manuscript 'Unveiling Dynamic Metabolic Signatures in Human Pluripotent and Neural Stem Cells' for review by PLOS Computational Biology. Your manuscript has been fully evaluated by the PLOS Computational Biology editorial team and in this case also by independent peer reviewers. The reviewers appreciated the attention to an important problem, but raised some substantial concerns about the manuscript as it currently stands. While your manuscript cannot be accepted in its present form, we are willing to consider a revised version in which the issues raised by the reviewers have been adequately addressed. We cannot, of course, promise publication at that time.

In particular, the reviewers state concerns regarding the model's goodness of fit and predictive capability, and thus  subsequent predictions/conclusions drawn from the model. In addition, a more in depth analysis of particular metabolites and pathways (and their dynamics) that comprise the metabolic signature in response to the change in nutrient concentration would strengthen the paper.

Sincerely,

Stacey Finley, Ph.D.

Associate Editor

PLOS Computational Biology

Jason Haugh

Deputy Editor

PLOS Computational Biology

[LINK]

Reviewer's Responses to Questions

**Comments to the Authors:**

Reviewer #1: Vasconcelos e Sa et al. have applied process dynamics and process control models to metabolic data from human pluripotent (hiPSC) and neural stem cells (hNSC) following a step change in extracellular glutamine concentration. This is a very interesting study because new approaches are indeed needed to understand the dynamic and tightly regulated nature of metabolism. The authors’ strongest findings are that 1) metabolic control is cell-type specific, 2) metabolites such as histidine and alanine can exhibit no differences in initial concentration but different dynamics upon glutamine step change (Figs. 5 & 6), and 3) fold-change data can better discriminate hiPSC and hNSC than either initial or final concentrations (Fig. 5A). The metabolomic data appear to be of high quality, and the authors have used a very interesting stirred-tank bioreactor system that appears to maximize biological reproducibility.

That said, there are several issues which need to be addressed before this manuscript would be acceptable for publication in PLoS Comp Biol. Primarily, the authors have failed to demonstrate 1) that their approach has revealed “key metabolic regulators involved in cell fate decisions” and 2) that their approach has predictive value (as opposed to simply being a nonlinear regression that can’t predict out-of-sample data).

Major points:

- The authors should include their data in a public repository (eg, MetaboLights) or provide the data in Supplemental Tables. It is unacceptable that the authors state “The datasets used and/or analysed during the current study are available from the corresponding author on reasonable request” (page 21).

- The authors fit their process control model to one glutamine step change (2.5 to 15 mM), but they have not used their model to make any predictions about the response to other perturbations. For example, can a model fit to the 2.5 to 15 mM step change data successfully predict the dynamics upon switching to a different glutamine step change (i.e., 2.5 to 8.75 mM)? This would demonstrate that their models have some general applicability and would argue that this is a meaningful modeling approach.

- Figure 5A: The authors show that “fold-change of steady-states” is better at discriminating hiPSC and hNSC (using ROC) than either initial or final concentrations alone. This certainly argues that steady-state analysis is important. However, the authors have not shown that dynamic process control modeling improves the ability to discriminate between cell types. Data showing that transient profiles of metabolic pools (or the parameters derived from process control models) provides an even stronger discrimination of the cell types would significantly strengthen the authors’ conclusion that their process control modeling approach is valuable.

- Figure 6C: the authors have identified metabolites with cell-specific dynamics (Fig. 6), but they have not provided any analysis that metabolites with cell-type specific dynamics are more important for cell fate or function than metabolites with non-specific dynamics. Such data would be essential for the authors’ conclusion that their approach has revealed “key metabolic regulators involved in cell fate decisions”. There is some discussion about amino acids and their important for hiPSC and hNSC (page 12), but there is no evidence here that amino acids and other cell-specific metabolites are **more important** than other non-cell-specific metabolites for stem cell phenotypes. The authors need to strengthen their analysis here or remove their substantially soften their claims that this approach reveals “key metabolic regulators”.

- The authors have provided no insight into what controls the cell-type specific differences. Is there gene expression or protein expression data that might explain why these cell-type specific differences occur?

Minor points:

- I was unable to access the Supplemental Tables. The link in the manuscript pdf took me to a word doc that described but did not actually contain the Supplemental Tables.

- Figure 2: have the authors done any false discovery rate (FDR) correction? The figure says “Adjusted P=0.05” but I could not find any description of whether or how the authors did FDR correction of their p-values.

- Figure 2: what time point is considered the “steady state”? It is not clear from the text or the legend what time point is used here. It is not clear how confident a reader should be in the authors’ assessment that these metabolites have reached steady state.

- The authors state that their model “successfully fitted at least 60% of the metabolites for each cell line” (page 8), but they have provided no discussion about why the model works for some but not other metabolites. Do the authors have any insight into why this modeling approach succeeds or fails for a given metabolite? Is there some shared characteristic among the fitted and not-fitted metabolites?

- Figure 4: it is unclear what part of this figure demonstrates the “mathematical modeling”. Is it the solid line in the figure? Probably, but the authors have not clearly stated that the solid line comes from their process control models.

- Figure 6: Are there classes / pathways of metabolites enriched in the identified metabolites with cell-specific or non-specific dynamics? The authors claim that the identification of amino acids is significant, but there is no statistical justification for this observation. The authors should attempt metabolite enrichment analysis or similar between these two classes of metabolites.

- The data for alanine and histidine is shown twice: Fig. 5B and Fig. 6D.

Reviewer #2: The authors of this manuscript describe metabolic responses of two human stem cell lines to step changes in glutamate availability. This dynamic, short-term response analysis of metabolite concentration is fairly novel and provides interesting insights into dynamics of metabolite pools. Furthermore, the use of control theory to model metabolite dynamics is new and potentially enlightening. Overall, the experiment is well-designed and metabolite analysis is also a strength. The manuscript is well-written and easy to follow. The simplicity of the study has both advantages and drawbacks. On one hand it provides a clear proof-of-concept, but it also leaves the reader with many unanswered questions. This work would be a valuable addition to stem cell metabolism literature as a proof-of-concept of the authors’ novel approach, but I encourage the authors to think about how additional analysis of system robustness and dynamics can be added to this study. Also, indications of how future studies will be designed would be valuable.

Specific comments:

1. My most significant comment involves a lack of pathway level analysis. While investigating individual pathways is useful, some analysis and discussion of how metabolites in the same pathway change (e.g. energetic and amino acid synthesis pathways) would go a long way to convincing the reader that this methodology can identify network dynamics accurately.

2. p. 5, last sentence. It would be useful to show data demonstrating the system reached a steady state by 2 hr in a supplemental figure.

3. System robustness should be discussed and investigated, at least in a modeling approach. For example, is the system reversible and how does it respond to step changes of different magnitude, or ramped changes (ie fed batch reactor). This analysis could be done for a subset of the metabolites and pathways of interest in the authors’ experimental data.

4. Overall the statistical analysis appears strong. I have one concern (p. 6, line 155). A t-test comparing two metabolites at 5% significance will determine whether 2 metabolites concentrations are different with 95% confidence. If I understand the test correctly, it’s not appropriate to use this to say two metabolites are not the same. The authors should clarify their test or perform a more appropriate test.

5. A discussion and/or data of how metabolic networks change would be helpful. Responses are quite rapid. How much of the response is the result of new enzyme synthesis vs. pathway flux changes utilizing existing enzymes?

6. Line 318-319. “…suggest that several amino acid pools are tightly regulated so that the homeostasis of cell phenotype is maintained.” This is an interesting idea but lacks support. It isn’t clear that the cells can be maintained long term after switching the metabolic environment. Also, more importantly, it isn’t clear that the amino acid pools are necessary to maintain cell state. I suggest moving this point to the discussion, and being more clear this is speculation.

Reviewer #3: The authors describe broad metabolic changes in response to short-term glutamine exposure in human iPSC and NSC, identifying changes in both steady state and dynamic (within 30 minutes) metabolic subgroups. In addition, they describe cell-type specific changes. It is well known that metabolic change occurs rapidly (long before changes in gene expression etc), so it is not surprising that upon glutamine challenge cells compensate by immediately modulating amino acid utilisation. Equally, it is not surprising that different cell types respond to the same challenge in a different manner, given that their baseline metabolic profile is different, and therefore have different metabolic needs. The physiological relevance of the challenge used is not clear, and overall the manuscript lacks sufficient context for the biological researcher to meaningfully interpret. This is significant given that the authors rationale for this work (abstract) being to contribute to understanding what cells need (as opposed to how they respond) given that media formulations are essentially based on tissue culture designed in the 1950s.

The manuscript requires editing by an English speaker as there are numerous errors.

Specific comments:

The authors state that metabolic changes are considered as ‘generally consistent’, thereby suggesting cell-type specific changes (although they do denote this as ‘cell-specific’ which is confusing). Based on their data, it is quite apparent that each cell line displays a vastly different response to the challenge (within cell-type). This is then made particularly apparent on line 218 where 99, 134, 116 and 71 metabolies were fitted to a mean fitting area of which only 60% were successful, as well as in Figure 6A where only 50% of metabolites are shared between hiPSC lines (although which metabolites this represents is not clear), and even fewer between the 2 NSC lines. As a result, the conclusions made are questionable, particularly those around preferred metabolic pathways (and the section on fitting metabolic profiles).

While the data would be of significant interest, no specifics are provided (i.e. details of actual metabolites changing beyond Figure 3 – which is steady-state, not the dynamics they suggest are key to understanding cell metabolism). For example, on line 373-5 of the discussion the authors note that ‘all metabolites with cell-specific dynamics and with decreased steady-state were amino acids. Some of the identified amino acids are known to have important cell regulatory functions…’. Yet no specific AAs are highlighted, which would be the key interest of those reading this paper. What metabolites common within a cell-type were fitted to their model (and in Figure 6A etc)? This makes it very difficult to reconcile with the discussion (suggesting that their method allowed an analysis of the essential and constant metabolic features of the cells), as well as the title (metabolic signatures) when no specific metabolites/signatures are divulged.

The authors state that hNSC alter their metabolism in response to glutamine by the ‘upstream section of the TCA cycle’ (line 184), but this is difficult to discern for anything other than lysine, and lipids (not TCA).

If a steady-state metabolic profile is reached within a short period of time, how do these profiles compare with previously reported profiles for each cell type? The authors state (line 365) that their results closely match data obtained by carbon-labelling. If these analyses have been performed, they should be included within the manuscript.

The use of the term ‘pluripotent’ implies both embryonic and induced pluripotent stem cells, however the authors have only analysed the latter, which are metabolically very different (as referenced in the paper – Panopoulos et al) and have also been documented to modulate metabolism in response to metabolite availability with varying abilities (which may relate to different methods of generation; I note the two lines used were generated either by retroviral or episomal reprogramming in the current submission). They should therefore not generalise to PSC throughout the manuscript (or in the title) and additional analyses should be performed to include examination of hESC. It would also be useful for the authors to comment within the manuscript about the differences between the cell lines, particularly in relation to differences in source/generation etc. What is the culture history (media used) for these lines?

What is the physiological relevance of using glutamine (and at the concentration specified) to challenge either cell type? Likewise, what, if anything, does the challenge do functionally to these cells? The authors could have assessed cell viability after the challenge (there is potential that cell death or differentiation is induced in the short window of time being examined, which may contribute to different metabolic changes seen between cell types), but more importantly, the authors should include measures of pluripotency/potency and differentiation capacity, as well as gene expression and epigenetics. Similarly, are cells still proliferating equivalently before and after the challenge? Without these latter analyses the authors can not conclude that their method is suitable for identifying the most effective phnotype modulator (line 387) the ‘identification of metabolites characteristics of a cell phenotype’ or ‘metabolic signatures of stemness’ (line 394).

Minor comments:

Line 54: in place of ‘in this last work’ simply state ‘Pavlova et al 2018’.

It is unclear how many biological replicates were performed, or whether the same stirred bioreactor culture was merely sampled on different days (i.e. a single biological replicate).

Please detail the ammonium accumulation analyses in the methods and results.

It is unclear whether excluded data points (lines 143-152) relate to metabolites or samples as a whole. If they relate to samples, then this particular sample should be removed from all analyses (not just certain metabolites where they are perceived as outliers).

Viability analysis should be quantified (%). The PI staining in Figure 1 is very difficult to visualise.

Figure 2: do the points on the graph represent samples (as a whole) or specific metabolites?

Line 236: Please expand on the specific transient dynamics that may contribute to distinct conserved metabolic features.

The authors need to cite the relevant stem cell metabolic literature (rather than studies on E.coli etc). Please also specify the cell types examined in cited literature (e.g. line 350-351 ‘previous works sharing related concepts or objectives focus on glycolysis and the PPP’ which does not relate to the cell types examined in the current study).

Ref 34 is used to suggest differences between cell types, however this reference relates more specifically to differences in metabolic features between embryonic and induced pluripotent stem cells (i.e. both are ‘pluripotent’ and therefore considered the same cell type) rather than differences between iPSC and the somatic cells from which they were generated.

Line 364-365: references should be provided to support this statement.

Line 415: Please include the passage numbers assessed and the incubator conditions used. Also, why were cells supplemented with Y-27632 when this is known to alter metabolism?

Please provide further details of the ‘Energy Metabolism Assay’ as this product no longer seems to be available (or is not produced by Biocrates).

It is unclear how much ‘randomness’ (i.e. variability) contributed to quantification of metabolites, and I am concerned as to why data were subsequently normalised. What is the precedence for using the normalisation equation?

The authors should provide a list of all (201) interrogated metabolites. Particularly with regard to 'energy metabolism' denoted in green in Figure 2.

In Figure 3, do the authors mean production and consumption for 'positive fold change' and 'negative fold change' respectively?

Figure 4: the authors use alanine as an example of a metabolite which maintains the same steady-state, yet it appears to shift quite dramatically in iPSC. This needs to be clarified.

Figure 6: Please define what the numbers in brackets (e.g. hiPSC 1 Set (37)) indicate. It is difficult to understand what 6B is depicting.

**Have all data underlying the figures and results presented in the manuscript been provided?**

Reviewer #1: No: The authors should include their data in a public repository (eg, MetaboLights) or provide the data in Supplemental Tables. It is unacceptable that the authors state “The datasets used and/or analysed during the current study are available from the corresponding author on reasonable request” (page 21).

Reviewer #2: None

Reviewer #3: Yes

PLOS authors have the option to publish the peer review history of their article (what does this mean?). If published, this will include your full peer review and any attached files.

Reviewer #1: No

Reviewer #2: No

Reviewer #3: No

---

## [Decision Letter · Decision Letter 1]

14 Jan 2020

Dear Dr Alves,

Thank you very much for submitting your manuscript, 'Unveiling Dynamic Metabolic Signatures in Human Induced Pluripotent and Neural Stem Cells', to PLOS Computational Biology. As with all papers submitted to the journal, yours was fully evaluated by the PLOS Computational Biology editorial team, and in this case, by independent peer reviewers.

The reviewers appreciated the attention to an important topic but identified some aspects of the manuscript that should be improved. In particular, one reviewer raises points regarding applying a false discovery rate correction and edits to the discussion section.

We would therefore like to ask you to modify the manuscript according to the review recommendations before we can consider your manuscript for acceptance. Your revisions should address the specific points made by each reviewer and we encourage you to respond to particular issues Please note while forming your response, if your article is accepted, you may have the opportunity to make the peer review history publicly available. The record will include editor decision letters (with reviews) and your responses to reviewer comments. If eligible, we will contact you to opt in or out.raised.

- Supporting Information uploaded as separate files, titled 'Dataset', 'Figure', 'Table', 'Text', 'Protocol', 'Audio', or 'Video'.

We hope to receive your revised manuscript within the next 30 days. If you anticipate any delay in its return, we ask that you let us know the expected resubmission date by email at ploscompbiol@plos.org.

Sincerely,

Stacey Finley, Ph.D.

Associate Editor

PLOS Computational Biology

Jason Haugh

Deputy Editor

PLOS Computational Biology

[LINK]

Reviewer's Responses to Questions

**Comments to the Authors:**

Reviewer #1: In this revised manuscript, Vasconcelos e Sa et al. have 1) improved availability of their metabolomic data; 2) softened claims that their approach has discovered “key regulators” of cell fate decision; 3) justified their choice of 2 h as steady state for this system.

The value of this manuscript still lies in the demonstration that fold-change of steady state metabolite concentrations are 1) cell-type specific (Fig. 2B) and 2) improve the ability to discriminate cell types (Fig. 5A). This data presents a valuable contrast to recent high-profile publications of steady state metabolite concentrations across 100’s of cell lines (e.g., PMID 31068703). I do, however, wish the authors had more thoroughly demonstrated either 1) that dynamic process control modeling improves the ability to discriminate between cell types or 2) that metabolites with cell-type specific dynamics are more important for cell fate or function than metabolites with non-specific dynamics.

That said, I think the manuscript would be acceptable for publication pending correction of these minor points:

Minor points:

- The authors state in their response that false discovery rate correction “was not performed as the goal of Fig2A was to give an initial indication of what metabolic pathways had increased or decreased metabolic levels, rather than identify specific metabolites. FDR is commonly used for larger datasets [3], for instance for gene expression data where specific genetic changes are searched for.” The cited reference [3] in the Response (PMID 26596774) investigates an alternative method of FDR correction (“local false discovery rate (lfdr)) but does not suggest that metabolomics data does not require FDR correction. The authors should either apply this lfdr approach or else do traditional FDR correction (e.g., Benjamini-Hochberg) to Figure 2A. It’s simply wrong to ignore the FDR correction for this data (which is smaller than gene expression data but still large enough to warrant FDR correction).

- The discussion about the role of amino acids in cell fate decision that is currently in the Results section should be moved to the Discussion because this is speculation based on the authors’ results, rather than actual Results (Line 314: “After all, amino acids are known to be involved in cell fate decisions. For instance, a controlled level of intracellular methionine and of the enzymes involved in its metabolism has been shown to be crucial for maintenance of pluripotency in hiPSC [25]. In a specific subtype of breast cancer with stem-cell like properties, asparagine has been pointed out as a key factor for governing metastasis. Besides that, serine supports proliferation of the breast cancer cells [26]. In addition, serine as well as glycine, are involved in one-carbon metabolism that influence epigenetics in human cells [27].”)

Reviewer #2: While preliminary in nature, the approach here warrants publication. The authors plan to follow up on this publication in subsequent work to address the model robustness, which will be crucial toward establishing advantages offered by their control theory approach.

Reviewer #3: The authors have adequately addressed the majority of concerns raised by all reviewers.

If phenotypic markers and cell viability have been assessed, it would be useful to include a note.to this effect within the manuscript.

How replicates were performed needs to be included in the manuscript. While it is appreciated that multiple stirred reactors would not be available, it needs to be clarified whether the replicates represent different thaws of the same cell line repeated (or just a single thaw resampled on different days) as it remains unclear. If the latter then this would not be a replicate.

My interpretation of the response to point 28 is that where data were lost for a given timepoint this was excluded, but the other.time points for the same sample were not. This is not apprppriate, particularly given the conclusions around dynamicism, as each timepoint should be represented by all samples. If not, then the changes over time will not necessarily reflect consistent change (especially when the n number is unclear).

I was not questioning the validity of the method in 29, rather that quantitative conclusions were being made from qualitative data. The issue with the visibility of the images for publication remain. I printed the images in colour, but they will not print well for those that only use B&W (nor do they reproduce well on screen for the reader to discern the staining).

Point 32. By 'specific' dynamics I was referring to metabolites.

ROCK inhibitor use has become standard, despite little examination of physiological effects. There are however data showing that ROCKi alters metabolism at 24-72h post passaging. What baseline data have the authors generated to confirm that they do not see this change?

**Have all data underlying the figures and results presented in the manuscript been provided?**

Reviewer #1: Yes

Reviewer #2: Yes

Reviewer #3: Yes

PLOS authors have the option to publish the peer review history of their article (what does this mean?). If published, this will include your full peer review and any attached files.

Reviewer #1: No

Reviewer #2: No

Reviewer #3: No

---

## [Decision Letter · Decision Letter 2]

8 Mar 2020

Dear Prof Alves,

We are pleased to inform you that your manuscript 'Unveiling Dynamic Metabolic Signatures in Human Induced Pluripotent and Neural Stem Cells' has been provisionally accepted for publication in PLOS Computational Biology.

Best regards,

Stacey Finley, Ph.D.

Associate Editor

PLOS Computational Biology

Jason Haugh

Deputy Editor

PLOS Computational Biology

Reviewer's Responses to Questions

**Comments to the Authors:**

Reviewer #1: The authors have sufficiently addressed my comments about FDR correction, and I recommend the manuscript for publication.

**Have all data underlying the figures and results presented in the manuscript been provided?**

Reviewer #1: Yes

PLOS authors have the option to publish the peer review history of their article (what does this mean?). If published, this will include your full peer review and any attached files.

Reviewer #1: No

---

## [Editor Report · Acceptance letter]

8 Apr 2020

PCOMPBIOL-D-19-01343R2 

Unveiling Dynamic Metabolic Signatures in Human Induced Pluripotent and Neural Stem Cells

Dear Dr Alves,

I am pleased to inform you that your manuscript has been formally accepted for publication in PLOS Computational Biology. Your manuscript is now with our production department and you will be notified of the publication date in due course.

With kind regards,

Matt Lyles
